# Telomere Length Changes in Cancer: Insights on Carcinogenesis and Potential for Non-Invasive Diagnostic Strategies

**DOI:** 10.3390/genes14030715

**Published:** 2023-03-14

**Authors:** Zuzana Holesova, Lucia Krasnicanova, Rami Saade, Ondrej Pös, Jaroslav Budis, Juraj Gazdarica, Vanda Repiska, Tomas Szemes

**Affiliations:** 1Geneton s.r.o., 84104 Bratislava, Slovakia; 2Institute of Medical Biology, Genetics and Clinical Genetics, Faculty of Medicine, Comenius University, 81108 Bratislava, Slovakia; 3Comenius University Science Park, Comenius University, 84104 Bratislava, Slovakia; 4Slovak Centre of Scientific and Technical Information, 81104 Bratislava, Slovakia; 5Department of Molecular Biology, Faculty of Natural Sciences, Comenius University, 84205 Bratislava, Slovakia; 6Medirex Group Academy, n.o., 94905 Nitra, Slovakia

**Keywords:** telomere length, telomerase, telomeric cfDNA, liquid biopsy, cancer, biomarker

## Abstract

Telomere dynamics play a crucial role in the maintenance of chromosome integrity; changes in telomere length may thus contribute to the development of various diseases including cancer. Understanding the role of telomeric DNA in carcinogenesis and detecting the presence of cell-free telomeric DNA (cf-telDNA) in body fluids offer a potential biomarker for novel cancer screening and diagnostic strategies. Liquid biopsy is becoming increasingly popular due to its undeniable benefits over conventional invasive methods. However, the organization and function of cf-telDNA in the extracellular milieu are understudied. This paper provides a review based on 3,398,017 cancer patients, patients with other conditions, and control individuals with the aim to shed more light on the inconsistent nature of telomere lengthening/shortening in oncological contexts. To gain a better understanding of biological factors (e.g., telomerase activation, alternative lengthening of telomeres) affecting telomere homeostasis across different types of cancer, we summarize mechanisms responsible for telomere length maintenance. In conclusion, we compare tissue- and liquid biopsy-based approaches in cancer assessment and provide a brief outlook on the methodology used for telomere length evaluation, highlighting the advances of state-of-the-art approaches in the field.

## 1. Introduction

Telomeres have been studied for decades, as they are related to important biological processes such as aging and the development of various diseases. As they play a crucial role in maintaining chromosome integrity, its deregulation and telomere length (TL) changes may lead to various age-related disorders, cardiovascular disease, or cancer development. Recent findings suggest cell-free telomeric sequences in circulation may also play regulatory roles in fine-tuning of the immune system [1]. Despite many recent advances in the study of telomeres and telomerase, numerous challenges and questions remain unexplored [2]. Understanding the role of telomeric DNA in carcinogenesis gives hope for new screening/diagnostic utility and cancer treatment strategies [3]. Several potential applications in the molecular assessment of human diseases have been suggested. However, the organization and function of telomeres in the extracellular milieu remain understudied.

Liquid biopsies involving the analysis of various biomolecules contained in the patient’s bodily fluids has become increasingly popular recently because of its undeniable benefits over conventional invasive methods [4]. This approach is gaining significant attention in clinical settings due to its potential to detect and monitor cancers at early disease stages [5]. The analysis of telomeres could enhance the existing assays, providing valuable information regarding cancer diagnosis, prognosis, and treatment response. Furthermore, it may contribute to a better comprehension of the mechanisms essential for the development and progression of various cancers.

In this paper, we summarize the mechanisms responsible for maintaining telomere length to better understand the biological factors affecting telomere homeostasis across different types of cancer cells. We focused on the studies looking into liquid biopsy-based telomere length analysis in cancer assessments. For early identification of telomere changes at a disease stage when the concentration of circulating tumor DNA is below the detection limit of standard methods, it is essential to apply highly sensitive methods. We therefore also provide a brief outlook on the methodology used for telomere length evaluation, highlighting the advances of state-of-the-art technologies in the field.

## 2. Telomere Structure and Homeostasis

Telomeres are protected by specialized nucleoprotein capping structures consisting of DNA and shelterin protein complexes (Figure 1). Human telomeric DNA is composed of tandem repeats (10–15 kb at birth) of double-stranded DNA nucleotide sequence 5′-TTAGGG-3′, and a final 3′ G-rich single-stranded overhang (150–200-nucleotide-long), linked by telomere-binding proteins [6]. The 3′ G-rich single-stranded overhang folds back and invades the homologous double-stranded TTAGGG region, forming a telomeric loop (T-loop). This higher-order structure provides protection to the 3′-end by sequestering it from recognition by the DNA damage response (DDR) machinery [7]. Although the length of telomeres differs among various chromosomes and species, their sequence is similar across all eukaryotes. This would indicate that telomeres are to a great extent a long-preserved and archaic structure with a major evolutionary role in the protection of genome integrity [8]. 

Telomere shortening can occur through two distinctive mechanisms [9]. The first one takes place because DNA polymerase is unable to replicate the 3′ end of the DNA strand fully, which causes the telomeres to physiologically shorten and lose approximately 30–150 bp with each cell division [10]. The second is caused by oxidative stress resulting from an imbalance between antioxidant defenses and reactive oxygen species (ROS) production [11] that leads to DNA damage and is considered the leading factor responsible for the remaining telomere loss [12]. High guanine content causes telomeres to become targets of oxidative stress through the formation of 8-hydroxy-2-deoxyguanosine (8-oxodG). 8-oxodG represents an important marker of oxidative damage causing accelerated shortening [13,14]. Since telomeric regions are known to be less efficient in damage repair of single-stranded DNA, the telomeres that contain said damage will not undergo full replication during the next cellular division, and the sequence downstream of the damaged region will be lost [15]. When telomeres erode to a critical length, cells become senescent and undergo morphological and genetic changes resulting in cell cycle arrest or apoptosis and loss of tissue function [16,17]. Senescent cells are known to also produce inflammatory mediators that affect the neighboring cells. This process leads to damage accumulation in tissues and organs. Therefore, as individuals grow older, they acquire a larger number of senescent cells which is also accompanied by a range of age-related pathologies [18]. Some cells overcome senescence by the acquisition of genetic mutations in the *p53* gene or other checkpoint proteins. As a result, they continue to proliferate, acquire immortality, and proceed to carcinogenesis [17,19].

To acquire replicative immortality, cancerous cells need to overcome the shortening of telomeres [17,20]. Most cancers maintain telomere length by activating telomerase, while 4–11% use a telomerase-independent alternative lengthening of telomeres (ALT) mechanism. Telomerase is a tightly regulated enzyme composed of telomerase reverse transcriptase (TERT) (the primary regulator of telomerase activity via its core promoter region and numerous binding sites), telomerase RNA component (TERC) containing the template for telomere replication [21], and accessory proteins. This complex is active during human embryonic development, particularly around the time of blastulation [22] in the germline and stem cells, but is much more passive or absent in most somatic cells [23,24,25]. Not only is the *TERT* expression regulated by transcription factors but it is also influenced by the epigenetic status [26,27]. As opposed to the canonical function of DNA methylation in gene silencing, researchers have demonstrated that *TERT* expression correlates positively with the levels of methylation at the *TERT* promoter region. At the same time, it correlates negatively with the level of gene coding sequence methylation [28,29]. It was previously believed that ALT was a characteristic unique to cancerous cells, but more recent studies also detected an ALT mechanism present in human placental cells in early stages of gestation [30] as well as in endothelial, stromal, and some epithelial non-neoplastic cells [31]. We therefore hypothesize that this mechanism occurs naturally under different physiological conditions that are at this point not fully understood.

As telomeres shorten, they can also modify gene expression at the level of transcription by the telomere position effect (TPE) [32,33]. Due to the presence of constitutive heterochromatin the expression of subtelomeric genes is repressed. Longer telomeres can form loop structures that are complementary to internal regions of the genome, which can be located several Mb away from telomere, leading to transcriptional repression, whereas with short telomeres, the looping is lost and genes can be transcribed, which is called TPE over long distances (TPE-OLD) [33].

Although telomeres were considered to be transcriptionally silent, it has been recently shown that mammalian telomeres undergo transcription into telomeric repeat-containing RNA (TERRA) (Figure 1). This long non-coding RNA participates in telomerase activity, heterochromatinization, and telomere length regulation that may be crucial to telomeric homeostasis and functions [34,35].

## 3. Telomeres in Health and Disease

Telomeres are protective DNA–protein structures at the end of chromosomes that prevent genome degradation and inappropriate activation of DNA responses [18]. They can serve as a potential indicator for disease susceptibility and cellular aging, predictors of mortality, or a possible target for a particular treatment [36,37]. Hence, proper telomere maintenance is crucial for ensuring healthy cellular function and preventing age-related diseases that could endanger human life. Telomere length is a dynamic marker that reflects not only genetic predispositions [38], but can be also affected by host age [39], sex [40], hormones [41,42], exogenous life factors (e.g., stress [43], nutrition [44,45], exercise [46], obesity and weight loss [47], paternal age [48], socio-economic status [49], alcohol dependence [50], tobacco smoking [51]), and environmental exposures (e.g., air pollution [52], UV radiation [36]).

### 3.1. Role of Telomeres in Non-Cancer Diseases

Telomeropathies are characterized by premature telomere shortening caused by mutations in genes involved in telomere maintenance or the DDR system. Based on the age of disease onset, severity, and type of organ systems affected, telomeropathies can be divided into three groups: (I) telomeropathies with very early onset and severe course (e.g., Revesz syndrome, Hoyeraal–Hreidarsson syndrome, and Coats plus disease); (II) dyskeratosis congenita (DKC); and (III) cryptic telomeropathy, a form of DKC with a later onset [53]. Currently, telomere testing is already available (e.g., RepeatDx Telomere Length Testing, SpectraCell’s Telomere Test, Cenegenics Telomere Testing, Life Length’s HealthTAV, etc.) and can provide informational value for selected at-risk individuals. The evaluation of telomere length has become an important tool for the screening of DKC, as the extent of telomere shortening correlates with its clinical manifestations [54].

Telomere shortening has been associated with several disorders, including type 1 and type 2 diabetes [55], cardiovascular disorders (e.g., increased risk of atherosclerosis [56] and development of coronary heart disease [57]), and neurodegenerative diseases (e.g., increased risk of Alzheimer’s and Parkinson’s disease [58,59] and progression of Huntington’s disease [60,61]). Shortening of telomeres can also contribute to the development of osteoarthritis and osteoporosis and their length in chondrocytes and peripheral blood cells could be a marker of disease progression [62]. Last, but not least, telomere length aberrations can result in genomic instability and elevated telomerase activity, leading to a potential cancer predisposition [63].

### 3.2. Role of Telomeres in Cancer

In cancerous cells, high telomerase activity overcomes the limitation of replication thus avoiding activation of the DNA damage signaling pathway [64]. Several studies have shown a correlation between telomere length and the risk of different cancers [65,66,67,68,69,70]. It has been proposed that up to 90% of human cancers reactivate telomerase via multiple genetic and epigenetic mechanisms that include *TERT* and *TERC* gene amplification, *TERT* promoter (*TERTp*) mutation and methylation, genomic rearrangement of *TERT* (e.g., insertion of active enhancers close to the *TERT* gene), post-transcriptional regulation by microRNAs, and alternative splicing of the *TERT* gene [67,68].

*TERTp* mutations are likely to be the most important mechanism participating in telomerase reactivation and *TERT* expression adjustment [68]. It represents the most common non-coding mutations in cancer [63]; however, since *TERTp* is typically not covered by the whole-exome sequencing (WES) analysis, it is absent from most genomic studies. Said mutations may not only affect *TERT* expression but also telomere length and telomerase activity by way of abrogating telomerase silencing [71]. Moreover, in cancerous cells, they are generally associated with higher levels of *TERT* expression. The two most frequent *TERTp* mutations are C228T and C250T transitions found at -124 bp and -146 bp, respectively from the site of the transcription start [72,73]. Such mutations create a new binding site for the E-twenty six (ETS) family of transcription factors, resulting in the increased *TERT* expression which was initially observed in melanomas [72] and later in other cancer types [74]. Somatic mutations of *TERTp* were detected at different frequencies in cancers with high (e.g., breast cancer 0.9% [75], testicular cancer 3% [76], intestinal cancer [77], AML and non-Hodgkin’s lymphoma [78,79]) and low (melanoma 64–80% [73], glioblastoma ~84% [74], bladder cancer ~65% [80], and hepatocellular carcinoma 32–45% [81]) proliferative potential [74]. *TERTp* mutations have not been detected in blood cell lines after in vitro culturing [82], and its detection was also negative in a cohort of patients with hematological malignancies, except for patients with mantle-cell lymphoma [83]. In cancers with low proliferative potential, *TERTp* mutations are recognized as a late event in tumorigenesis [74]. Moreover, in other cancers, e.g., basal cell carcinoma, *TERTp* mutations may develop as a result of exogenous factors, e.g., ultraviolet radiation or carcinogens of chemical origin, and are therefore recognized as an early tumorigenic event [84]. *TERTp* mutations are believed to restore the shortest telomeres which extends the life span of the affected cells; however, they fail to avert telomere shortening in general. Critically short telomeres lead to genomic instability, resulting in further increase in telomerase expression which is necessary for continued cell proliferation [85]. An intriguing aspect of *TERTp* mutations is the possible cooperation with mutations of other genes of high significance in tumorigenesis, e.g., *BRAF*, *FGFR3*, and *IDH* [86,87,88].

Several SNPs in the *TERT* locus have been associated with cancer risk [84]. The most intensively studied polymorphisms associated with multiple types of cancers are rs2736100 and rs2736098. Meta analysis conducted by Zhang et al. [89] showed association between rs2736100 and an increased risk of thyroid cancer, bladder cancer, lung cancer, glioma, and myeloproliferative neoplasms. Further still, rs2736098 was shown to increase the risk of bladder and lung cancer, while both variants were associated with a decreased risk of breast and colorectal cancer and an increased cancer risk across all studied populations [84,89]. Moreover, Tian et al. [90] found 13 variants within the *TERT-CLPTM1L* 5p15.33 chromosomal region associated with susceptibility to 11 various cancers and 1 non-cancer disease (idiopathic pulmonary fibrosis).

On the other hand, ALT is frequently observed [25,29] in a subset of neoplasms mainly represented by sarcomas and gliomas with a lack of telomerase activity. This homologous recombination-based process involves synthesis of new telomeric DNA using a DNA template, where the telomere can either use itself or a telomere on a sister chromatid or even another chromosome as a template. This is the reason why cells using this mechanism are characterized by long and heterogeneous telomeres [91].

## 4. Applications of Telomeric DNA in Cancer Assessment

Regulation of telomerase expression, assembly, and function occur across all levels of gene expression and is governed by a range of intracellular and environmental stimuli that share substantial overlap with pathways that control malignant transformation and tumor progression [92]. Thus, telomere length is an important indicator of tumor progression and survival in cancer patients [70,93]. However, studies reported inconsistent findings [94] reflecting the complex role of telomeres in cancers. The diversity of cancer types, ethnicities, study designs, measurement methods, and selected tissues for telomere length measurements in previous papers further complicates the observed association. Given the severe global burden of various cancers [95], reaching a better understanding of the association between telomere length and cancer could deliver highly important insights into how to improve the prevention and treatment strategies for various cancers. The diverse role of telomeres in different types of cancers has been revealed, but further validations within large-scale prospective studies and a more detailed investigation into the underlying biological mechanisms will, no doubt, be necessary [94].

Liquid biopsies have shown tremendous potential in detecting early-stage cancers, selection of targeted therapies, monitoring disease progression, or post-surgical detection of residual disease [96]. This is enabled through analysis of circulating tumor cells, extracellular vesicles, extracellular nucleic acids and proteins, etc. [97]. Cell-free DNA (cfDNA), fragments shed by cells during apoptosis, necrosis, or secretion, is considered to be the most promising biomarker [98]. Since most tumors are in contact with blood, liquid biopsy usually involves blood sampling, although other body fluids can also be analyzed [99]. cfDNA shows several advantages over conventional cancer screening tools: (I) detection of early-stage cancer; (II) simultaneous detection of multiple cancers by providing information on the tissue of origin; (III) blood sampling is safer, easier, more tolerable, and less invasive than traditional screening methods; and (IV) the potential to depict the entire molecular and (epi)genetic landscape, irrespective of intratumor heterogeneity [100].

The presence of cfDNA in circulation is not exclusive to cancer, but the levels in cancer patients are increased due to its additional release by tumor cells, known as circulating tumor DNA (ctDNA) [101]. A small amount of ctDNA in the large cfDNA pool [102] and somatic mosaicism in plasma remain an immense challenge for accurate interpretation of liquid biopsy results. The accumulation of somatic mutations and clonal expansion of hematopoietic stem cells is part of normal aging. The detection of such clonal hematopoiesis (CH) mutations has been repeatedly shown to be a source of the biological background noise of liquid biopsies performed from blood samples. Faulty classification of CH mutations as tumor-derived mutations may even result in inappropriate management of therapy [103]. Thus, a strategy (ultrasensitive targeted sequencing of matched cfDNA and white blood cells before and after chemotherapy/surgery) to distinguish ctDNA alterations from CH-related cfDNA mutations should be applied [104].

Single liquid biopsy biomarkers often cannot deliver accurate prediction of the disease state due to heterogenous phenotype and disease expression across individual patients. To deal with this challenge, researchers combine multiplexed measurements spanning across a full range of various biomarkers that combine to define robust signatures for different disease states. Machine learning may be regarded as a helpful instrument delivering automated discovery and detection of said signatures, especially as new technology leads to increasing quantities of molecular data output [105].

### 4.1. Tissue/Cell-Derived DNA-Based Approaches

The measurement of the leukocyte telomere length (LTL) from peripheral blood represents the gold standard for evaluating the dynamics of TL changes. It is usually provided by way of qPCR [106] (other methods described in the following sections are also used). In most cancer diagnostic research articles, LTL is used as a reference to represent other tissues [65] to compare TL changes in the tissues of interest [107]. To explore the extent to which telomere length in whole blood reflects TL in other tissues, Demanelis et al. measured relative telomere length (RTL) in more than 25 tissues from 952 donors [108]. Considering the differences between the tissue types, the relationship between telomere length and cancer risk was debatable because of the dual role of telomeres in carcinogenesis [109]. Both shortening and elongation of telomeres could increase cancer risk. Cells with longer telomeres have higher proliferative potential and are more prone to acquire mutations. On the other hand, while telomere shortening is considered a protective mechanism against tumorigenesis, short telomeres lead to increased genomic instability that promotes carcinogenesis [65].

The research conducted by Lansdorp showed that the differences in TL of individual white blood cell types increase with advancing age (depending on the number of cell divisions) compared to hematopoietic stem cells [110]. To achieve more reliable results, some authors prefer measuring telomere lengths in a specific type of white blood cells, e.g., mononuclear cells [111], granulocytes [112], or lymphocytes [113,114]. A meta-analysis by Adam et al. summarized the results of 61 studies (a total of over 11.6 thousand cases with solid tumors and over 3 thousand cases with hematological malignancies) analyzing LTL, as well as TL changes in tumors and their surrounding tissue. The study concluded that a longer TL predicted a better prognosis for patients with chronic myeloid leukemia and urothelial cancer, but TL did not play any prognostic role in the other malignancies that were examined [115]. In another large-scale meta-analysis, Zhu et al. examined the association between TL and overall cancer risk in population studies. Having included over 23 thousand cases and almost 69 thousand controls, the study concluded a non-significant association between short TL and overall risk of cancer, excluding tumors of the digestive system, where they found a significant dose–response relationship [69]. An extensive genome-wide association study (GWAS) conducted on over 823 thousand participants of a United Kingdom biobank suggested that shortening of TL contributes to skin aging but reduces the risk of skin melanoma and non-melanoma skin cancer [116]. In a large-scale study conducted on more than 420 thousand patients and almost 1.1 million controls from the NHGRI-EBI GWAS catalog, it became evident that lengthening of TL due to germline genetic variants was generally associated with an increased incidence of certain cancers. The strongest association was found for glioma, low-grade serous ovarian carcinoma, lung adenocarcinoma, neuroblastoma, bladder cancer, melanoma, testicular cancer, kidney cancer, and endometrial carcinoma [117] (Table 1). The results of large-scale studies differ in some cases, while the authors attribute this diversity to the different sample processing methods [109,118] or statistically insignificant patient group sizes [119,120,121].

The advantage of measuring LTL is the simplicity and availability of blood sampling that can be performed by a nurse in an outpatient setting, while measuring TL from a tissue biopsy requires a team of health workers cooperating during invasive sample collection and presents certain health risks for the patient. However, it is necessary to thoroughly understand the correlation between LTL and TL in tumor tissues in the respective types of cancer in order to use LTL as a biomarker for monitoring the dynamics of cancer development and treatment.

### 4.2. Cell-Free DNA-Based Approaches

Over the past decade, much effort has been devoted to the development of non-invasive approaches for cancer diagnosis and monitoring to replace the conventional invasive techniques [129,130,131]. Although associations between TL and cancer risk or prognosis have been extensively investigated in tumor tissues [69], cell-free telomeric DNA (cf-telDNA) remains poorly explored despite its potential suitability to become an informative genetic biomarker for many cancers (Table 2). The analysis of cfDNA for clinical diagnostic applications remains challenging since a number of methodological and pre-analytical factors (e.g., biological variables and sample handling) limit its clinical sensitivity in body fluids [132]. Zinkova et al. determined RTL and estimated the total amount of telomeres in plasma, serum, and whole blood. Serum has a higher content of telomeric sequences than plasma due to its larger amounts of total cfDNA. However, plasma provides higher values of the RTL ratio in comparison to serum, suggesting the different modes of physiological processing of cfDNA in these biofluids [1].

Wu and Tanaka quantified plasma cf-telDNA levels in breast cancer patients [140]. They found that its levels were significantly reduced in patients with no prior treatment when compared to control individuals, suggesting that plasma cf-telDNA levels are associated with breast cancer susceptibility [139]. In gastric cancer, shortened telomeres were associated with gastric cancer risk [134], while RTL was found to be significantly lower in endometrial cancer patients [135]. This data is in accordance with the existence of paradoxically shorter telomeres in patients with cancer. The study by Urfali et al. compared the TL of cfDNA with the length determined in peripheral blood and tumor tissues in 40 patients with five different types of cancer (breast, colon, stomach, lung, and rectum). The plasma TL of the cancer group was significantly longer than that of the control group [138]. Similarly, the serum telomeres were reported to be longer in patients with both cirrhosis and hepatocellular cancer (HCC) [136,143], and the same trend was observed in patients with chronic hepatitis B infection who developed HCC after 5 years of follow-up [137]. As telomeric G-tail DNA cannot be detected with conventional sequencing technologies, Zheng et al. developed a high-preservation technology to quantify telomeres in cfDNA. They showed that short G-tail DNA was nearly 20-fold higher in HCC patients than controls and was strongly correlated with lower survival [142]. A study by North indicated that serum cf-telDNA was present in glioblastoma patients at nearly double the amount of the levels in the non-cancer control group. The increased presence of telomeric sequences in serum was directly correlated with glioblastoma disease conditions and thus may represent a useful biomarker for treatment response and measurement of tumor burden [141]. Terminal restriction fragment (TRF) analysis in plasma has shown telomere shortening in ovarian cancer patients compared to non-cancer controls, while TL correlated with the patients’ clinical status. Circulating TRF was in the normal range in long-term (10-year disease-free) survivors, indicating that short (tumor) TRF had been replaced by longer (normal) TRF; therefore, further assessment will be necessary to confirm the value of TRF analysis in cancer diagnosis or monitoring of treatment efficacy [133].

TERT is a key rate-limiting catalytic subunit with low or absent expression in normal cells. However, it plays a key role in sustaining the unlimited replication potential of cancerous cells [20], suggesting that TERT expression levels may represent a specific tumor development biomarker [144]. Consistent with this crucial role in pathogenesis, circulating cell-free *TERT* mRNA may be detected in the plasma of cancer patients. Moreover, the measured levels significantly correlate with those in tumor samples [145]. Further still, cell-free *TERT* mRNA is not detected in the plasma sampled from healthy individuals [145,146]. In particular, several studies have shown that circulating *TERT* mRNA is an independent prognostic marker for various tumor types [147], including gastric [148], prostate [149], lung [150], and colorectal cancers [145,151,152]. Furthermore, TERT mRNA levels found in the plasma of patients suffering from rectal cancer have been identified as a predictive marker for response to therapy [151,152,153]. Cangemi et al. [154] found that cancer patients showed considerably higher levels of circulating *TERT* mRNA as opposed to individuals from the non-tumor cohort both not only at baseline but also in follow-up samples. These findings are in line with the evidence that an increased *TERT* expression represents a hallmark of cancer [20].

## 5. Methodological Aspects of Telomere Length Measurement

Because of the importance of telomeres in human physiology, various methods have been developed to measure telomere length [155,156]. Several of them are well described and compared in previous review papers, while the most frequently used ones are listed below: ranging from the gold standard TRF method [157,158] through quantitative PCR (qPCR) for measuring the ratio of telomere repeats to single copy gene copy numbers [106,159], various fluorescence in situ hybridization (FISH) techniques (e.g., Q-FISH, Flow-FISH) [160,161], Single Telomere Length Analysis (STELA) for telomeres amplification and length measurement by gel electrophoresis [162], Telomere Shortest Length Assay (TeSLA) detecting the shortest telomeres by combining ligation, PCR-based methods, as well as Southern blot analysis [163], to massively parallel sequencing (MPS). However, the repetitive nature of telomeric sequences complicates its analysis by short-read MPS due to frequent mapping errors. Thus, sophisticated bioinformatic tools for MPS-based analysis of telomere content and composition are required.

Castle et al. [164] provided a proof of concept to measure telomere content by MPS in 2010. The prominent feature of this technique is the bioinformatic processing of the sequencing data by suitable tools (Figure 2). Motif_counter is a simple bash script that allows the scanning of all reads for a specific repeating motif sequence [165]. The first dedicated tool to estimate telomere length rather than just telomere content was TelSeq [166]. CompuTel takes a different approach to identify telomere reads by aligning each of the relevant reads to a hypothetical telomere reference [167]. The first tool designed specifically to estimate mean telomere length from WGS data is Telomerecat. The method does not depend on the number of present telomeres, making it suitable for the purpose of estimating telomere length in cancer studies [168]. TelomereHunter is another tool designed to analyze telomeres in WGS data [169]; however, it reports an estimate of relative telomere content rather than telomere length. The main advantage of another tool, qmotif, is its quick application, while its output correlates strongly with the output of the other mentioned tools [170].

The PubMed search for the term “telomere length” AND “cancer” has shown a gradual decrease in the number of publications from 2017 onwards, suggesting that scientists have depleted the potential of conventional approaches to studying telomere length in cancer (Figure 2). Thus, utilizing state-of-the-art techniques will be beneficial to achieve the full potential of telomere biology in cancer assessments [171].

Long-read sequencing enables the sequencing of DNA longer than 10 kilobase pairs, representing an emerging approach suited for the study of long repetitive elements like telomeres [172]. Indeed, the technique proved to be a game-changer, as it has been recently applied to create the first complete reference sequence of a human genome [173]. The advent of long-read sequencing has provided new opportunities to resolve the length and sequence content of telomeres [174]. Thus, fundamental progress in the study of telomere dynamics can be reasonably expected in the near future.

## 6. Conclusions

To our knowledge, there is no accepted consensus on whether telomeres are lengthened or shortened in cancer. The literature indicates that there are telomere length differences between diseases, cells/tissues, types of biological material, and methods used for telomere analysis. Since the current knowledge is based on various methodologies, understanding the strategies for telomere length measurements is necessary to compare the results published in different studies. Despite the increasing amount of next-generation sequencing data entering the clinical routine, analyses of telomere length by short-read MPS have not yet been widely implemented in practice. Another factor coming into play is the bioinformatics tools enabling telomere detection or telomere length measurements that have reached a high level of applicability to estimate telomere length from MPS data. Telomere length testing has the potential to provide assistance to physicians when identifying patients who are at risk of developing telomere-related disorders and their associated complications. However, the bilateral nature of telomere length changes in cancer remains a challenge to be addressed. It will therefore be necessary to explore other biological factors that could explain this phenomenon. The current methodology seems to be sufficient, but large-scale validation studies will be necessary to confirm cf-telDNA length as a reliable biomarker for biomedical applications in cancer screening, early diagnosis, or monitoring.

## Figures and Tables

**Figure 1 genes-14-00715-f001:**
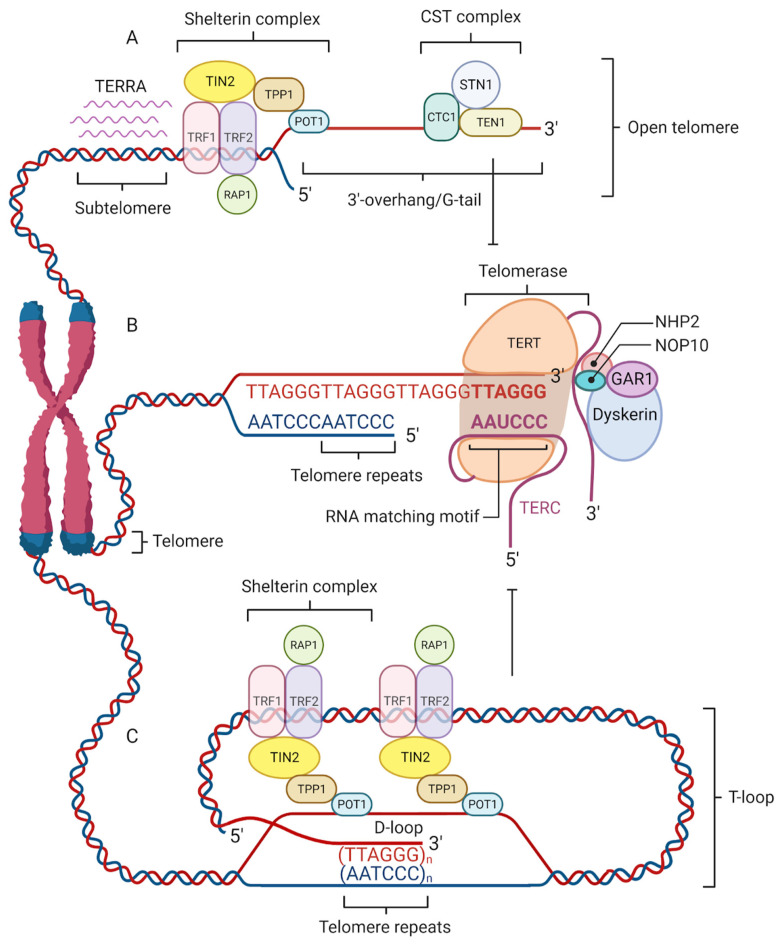
Four factors contribute to telomere maintenance: the shelterin complex, telomerase, telomeric repeat-containing RNA (TERRA), and the CST complex. The shelterin complex protects telomeres and regulates telomere elongation. TRF1/2, RAP1, and TIN2 are associated with double-stranded DNA; POT1 and TPP1 bind to the single-stranded G-tail DNA and are responsible for recruiting telomerase to telomeres (**A**). The shelterin complex coordinates the T-loop (Telomere loop) formation into which a 3′-overhang extends to form a small D-loop (Displacement-loop) and protects the end of the chromosome from damage (**C**). The absence of a shelterin complex causes telomere uncapping and thereby activates damage-signaling pathways. Telomerase is a reverse transcriptase enzyme that carries its own RNA molecule, which is used as a template in telomere elongation. This ribonucleoprotein complex consists of TERT (telomerase reverse transcriptase), TERC (telomerase RNA component), dyskerin, NOP10, NHP2, and GAR1 (**B**). TERRA is transcribed from telomere DNA and together with the shelterin complex inhibits telomere lengthening by telomerase (**A**). The CST complex localized on the single-stranded 3′ overhang prevents telomerase from binding to the 3′-overhang and interacts with DNA Polα-primase during telomere replication (**A**).

**Figure 2 genes-14-00715-f002:**
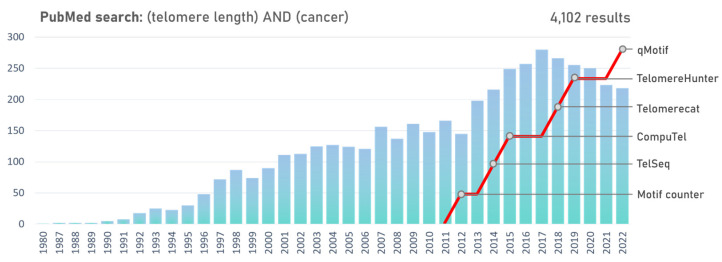
Number of publication records in the PubMed database for the keywords “telomere length” AND “cancer”. The red line represents the cumulative count of bioinformatic tools dedicated to telomere length analysis.

**Table 1 genes-14-00715-t001:** Summary of studies evaluating telomere length across various types of cancer. Our review includes a total number of 502 studies ^1^, and data from the UK biobank, TCGA, and NHGRI-EBI GWAS catalog that, together, analyzed 3,396,477 samples ^1^ of patients with various cancers, other health-related conditions ^2^, and control individuals.

Type of Cancer/Condition	Material ^3^	Method	No. of Studies/ Source	No. of Patients/Controls	Alt. ^4^	Ref.
breast	blood, tissue	dot blot, FISH, qPCR, Southern blot, STELA	36	6311 cases	↓	[118]
facial skin aging, NMSC, skin melanoma	blood	qPCR	UK biobank data	451,444 cases/ 372,016 controls	↓	[116]
APML, Barrett carcinoma, breast, CLL, CRC, Ewing sarcoma, esophageal, gastric, gastroenteropancreatic, glioblastoma, glioma, HCC, head and neck, HL, low-risk B-cell precursor ALL, neuroblastoma, NSCLC, ovarian, prostate, renal, urothelial	blood, tissue	dot blot, FISH, MMqPCR, qPCR, Southern blot, STELA	61	14,720 cases	↑	[115]
bladder, breast, CRC, endometrial, esophageal, gastric, glioma, head and neck, hematological, liver, lung, melanoma, multiple myeloma, NMSC, non-HL, ovarian, pancreatic, prostate, renal, soft tissue sarcoma, thyroid	blood	qPCR	112	64,184 cases/ 278,641 controls	↑	[122]
bladder, breast, CLL, CRC, esophageal, gastric, glioma, head and neck, HCC, HL, lung, ovarian, prostate, renal, skin basal cell carcinoma, skin melanoma, other conditions ^2^	blood, tissue	NA	21 meta- analyses	153,451 cases/ 180,130 controls	↓	[123]
melanoma	blood	qPCR	1	970 cases/ 733 controls	↓	[124]
prostate	blood, tissue	FISH, qPCR	12	2130 cases/ 2131 controls	↓	[125]
bladder, brain, breast, CRC, endometrial, hematological, liver, lung, melanoma, pancreatic, prostate, renal, skin-basal cell carcinoma, skin-squamous cell carcinoma	blood	qPCR	25	13,894 cases/ 71,672 controls	↑	[109]
NSCLC, SCLC	blood, tissue	qPCR, Southern blot, TRF	14	1503 cases	↑↓	[121]
CRC	blood	qPCR	7	4951 cases/ 7993 controls	↑↓	[126]
soft tissue sarcomas (ALT+/ALT−)	tissue	APB evaluation, FISH, TRF	8	551 cases	↑ ^5^	[91]
CRC	blood, tissue	qPCR	7	956 cases	↑↓	[120]
adrenocortical, ALL, bladder, breast, cervical, cholangiocarcinoma, CRC, diffuse large B-cell lymphoma, endometrial, esophageal, gastric, glioblastoma, glioma, HCC, head and neck, kidney chromophobe, lung, lymphoid neoplasm, ovarian, pancreatic adenocarcinoma, paraganglioma, pheochromocytoma, prostate, renal, sarcoma, skin melanoma, testicular germ cell, thymus, thyroid carcinoma, uterine carcinosarcoma, uveal melanoma	blood, tissue	WGS, low-pass WGS, WES, TelSeq	TCGA data	6835 cases/ 11,595 controls	↑↓	[29]
NMSC, skin melanoma	blood	LTL (qPCR)	8	3068 cases	↓	[127]
ALL, AML, APML, astrocytoma, brain stem tumor, breast, clear cell sarcoma, CLL, CML, CNS tumors, CRC, esophageal, Ewing sarcoma, gastric, germ cell, HCC, head and neck, hepatoblastoma, HL, lung, lymphoma, multiple myeloma, neuroblastoma, non-HL, ovarian, prostate, rhabdomyosarcoma, severe aplastic anemia, thyroid, Wilms’ tumor	blood, tissue	Flow-FISH, Q-FISH, qPCR, TRF	25	2261 cases	↑↓	[119]
B-cell lymphoma, basal cell carcinoma, bladder, breast, CRC, endometrial, esophageal, glioma, HCC, head and neck, gastric, lung, melanoma, myeloma, non-HL, oral cavity, oropharyngeal, ovarian, pancreatic, prostate, renal, skin melanoma, squamous cell carcinoma	blood, tissue	Flow-FISH, Q-FISH, qPCR	51	23,379 cases/ 68,792 controls	↓	[69]
AML, bladder, breast, CLL, CRC, esophageal, gastric, glioblastoma, glioma, HCC, head and neck, myeloproliferative neoplasms, neuroblastoma, NSCLC, ovarian, prostate, renal	blood, tissue	FISH, qPCR, Southern blot	33	15,722 cases	↓	[66]
lung	blood, sputum, tissue	Q-FISH, qPCR	9	2925 cases/ 2931 controls	↓	[128]
bladder, breast, CRC, esophageal, gastric, lung, non-HL, ovarian, prostate, renal, skin	blood, sputum, tissue	Q-FISH, qPCR, Southern blot	21	11,255 cases/ 13,101 controls	↓	[93]
bladder, breast, CRC, esophageal, Ewing sarcoma, gastric, glioblastoma, glioma, head and neck, leukemia, liver, lung, lymphoma, myelodysplastic syndromes, neuroblastoma, oral cavity, ovarian, prostate, renal	blood, tissue	dot blot, flow-FISH, qPCR, Southern blot, STELA, Telo assay	51	14,464 cases	↓	[70]
bladder, breast, CRC, endometrial, esophageal, glioma, head and neck, lung, neuroblastoma, ovarian, pancreatic, prostate, renal, skin, testicular germ cell, other conditions ^2^	blood, tissue	NA	NHGRI-EBI GWAS catalog	420,081 cases/ 1093,105 controls	↑	[117]
basal cell, bladder, bone, brain, breast, cervical, CLL, CML, CRC, endometrial, esophageal, eye and/or adnexal, gastric, HL, larynx, leukemia, lung, lymphoma, malignant melanoma, multiple myeloma, NMSC, non-HL, oral cavity, ovarian, prostate, renal, sarcoma/fibrosarcoma, skin, small intestine, squamous cell, testicular, throat, tongue, thyroid	blood	NA	UK biobank data	78,582 cases	↑	[94]

Abbreviations: acute lymphoblastic leukemia (ALL); alternative lengthening of telomeres (ALT); acute myeloid leukemia (AML); ALT-associated promyelocytic leukemia bodies (APB); acute promyelocytic leukemia (APML); chronic lymphocytic leukemia (CLL); chronic myeloid leukemia (CML); colorectal cancer (CRC); genome-wide association study (GWAS); hepatocellular carcinoma (HCC); Hodgkin’s lymphoma (HL); leukocyte telomere length (LTL); Not available (NA); non-melanoma skin cancer (NMSC); non-small cell lung cancer (NSCLC); small cell lung cancer (SCLC); single telomere length analysis (STELA); The Cancer Genome Atlas (TGCA); terminal restriction fragment (TRF); whole-exome sequencing (WES); whole-genome sequencing (WGS). ^1^ cases/controls may be duplicated within the cited studies; ^2^ detailed list of disorders is available in Appendix A; ^3^ tissues include: tumor tissue, normal tissue, surrounding mucosa, tumor associated stroma cells, biopsy slides, buccal mucosa, bone marrow; blood includes: whole blood, buffy coat, leukocytes, mononuclear cells, polymorphonucleocytes, granulocytes, lymphocytes, EBV-transformed lymphocytes, CLL samples, reference from B cells, and plasma DNA; ^4^ telomere shortening (↓)/lengthening (↑); conclusion of the study may not apply to all described diseases, more details in Appendix A; ^5^ increased in ALT+ patients.

**Table 2 genes-14-00715-t002:** Telomeric cfDNA alterations in telomere length/levels for various types of cancer.

Cancer Type	Body Fluid	Method	Telomere Alteration ^1^	No. of Patients/Controls	Ref.
ovarian	plasma	TRF	↓	32 + 10 ^3^/45	[133]
gastric	serum	qPCR	↓	86/86	[134]
endometrial	serum	qPCR	↓	40/31	[135]
hepatocellular ^4^	serum	qPCR	↑	140/280	[136]
hepatocellular ^4^	serum	qPCR	↑	37/286	[137]
breast (16), colon (16), stomach (3), lung (3), and rectum (2)	serum	qPCR	↑	40/20	[138]
breast	plasma	qPCR	↓ ^2^	28/28 47/42	[139] [140]
glioblastoma	serum	qPCR	↑ ^2^	40/9 ^5^	[141]
hepatocellular	plasma	BLESSING/ Telecon ^6^	↓ ^2^	60/40 ^7^ 63/50 ^8^	[142]

^1^ shortening (↓)/lengthening (↑); ^2^ decreasing (↓)/increasing (↑) levels; ^3^ long-term survivors; ^4^ patients with chronic hepatitis B infection; ^5^ controls with epilepsy; ^6^ WGS and machine learning approach to quantify short telomere G-tail DNA; ^7^ discovery phase; ^8^ validation phase.

## Data Availability

No new data were created or analyzed in this study. Data sharing is not applicable to this article.

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
