# Peer review of "Telomere Length Changes in Cancer: Insights on Carcinogenesis and Potential for Non-Invasive Diagnostic Strategies"

_genes, 2023, doi:10.3390/genes14030715_

Round 1

Reviewer 1 Report

This manuscript reviewed the structure of telomeres and mechanisms of telomere length changes, especially the applications of telomeric DNA in cancer assessment. I have 2 minor suggestions as follows.

1. Page 4, 3.1 Role of telomeres in cancer. It seems like the 3.2 or/and 3.3 parts are missing; otherwise, it's not necessary to use the subtitle.

2. Page 3, 3. Telomeres in health and disease summarised telomere shortening is associated with quite an amount of human diseases, which seems not related to the cancer theme.

Reviewer 2 Report

The current manuscript examined the association of telomere lengthening/shortening, particularly with cancer and other diseases. In addition, the limitations of the relationship between telomere length and diseases methodologically were mentioned. Overall, the role of telomere length in cancer diagnosis remains questionable, but the current manuscript has detailed this with pros and cons. However, some corrections to the article need to be made in order to increase readability. Telomerase and biomarker keywords are not mentioned in the abstract. Emphasis should be placed on these keywords in the abstract. Some abbreviations need to be explained in the first place they are mentioned. In addition, minor corrections regarding the language rules are required in the article. The corrections mentioned above are highlighted in the document.

Reviewer 3 Report

The article by Holesova et al. is a review entitled “Telomere Length Changes in Cancer: Insights on Carcinogenesis and Potential for Non-Invasive Diagnostic Strategies”, to be published in a special issue of Genes entitled “DNA Damage and Repair at the Crossroad with Telomeres”. This review aims to better understand the biological factors affecting telomere homeostasis in different types of cancer and to do so examines the mechanisms responsible for telomere length maintenance. In addition, this review also compares tissue and liquid biopsy-based approaches in cancer assessment and provides a brief outlook on the methodology used for telomere length evaluation, highlighting the advances of current state-of-the-art technologies in the field.

Major comments:

* This review gives a pretty good and recent picture of the current knowledge on the subject. All chapters have been well organized and documented. This was a challenge for the authors because there are really many data published in the literature. In the end, the review is relatively easy to read in spite of the huge amounts of data presented.

* A major improvement might be to construct a Figure providing the main characteristics in composition and organization of the human telomere, going with text around lines 60 to 120 (t-loop, 3’-overhang, shelterin, CST, telomerase, etc…), with or without ALT (?). Indeed, this would be a nice introduction, very useful for many clinician readers that sometimes do not know exactly where telomeres are located and how they function in terms of proteins and DNA.

Minor comments:

* Line 163: The abbreviation “WES”, for “whole exome sequencing” should be given here, without waiting it appears in one of the footnotes of Table 1.

Please check whether this is the case for other abbreviations.
